# Photodynamic Properties of CdSe/CdS Quantum Dots in Intracellular Media

**Thanh Binh Nguyen** [1], **Thi Bich Vu** [1,2,*], **Dinh Cong Nguyen** [1], **Thi Thao Do** [3], **Hong Minh Pham** [1] and **Marilou Cadatal-Raduban** [4]

[1] Institute of Physics, Vietnam Academy of Science and Technology, Hanoi 100000, Vietnam; tbnguyen@iop.vast.ac.vn (T.B.N.); ndcong@iop.vast.ac.vn (D.C.N.); phminh@iop.vast.ac.vn (H.M.P.)

[2] Institute of Theoretical and Applied Research, Duy Tan University, Hanoi 100000, Vietnam

[3] Institute of Biotechnology, Vietnam Academy of Science and Technology, Hanoi 100000, Vietnam; thaodo@ibt.ac.vn

[4] School of Natural and Computational Sciences (SNCS), Massey University, Albany, Auckland 0632, New Zealand; m.raduban@massey.ac.nz

[*] Correspondence: vuthibich@duytan.edu.vn; Tel.: +84-9635-91052

**Abstract:** CdSe/CdS quantum dots (QDs) were seeded into Jurkat cells using polyethylene glycol (PEG-1500) at different treatment times. Fluorescence microscopy images show that some QDs stick to the surface of the cells, while others appeared to be inside the cells. As it is difficult to ascertain whether the QDs are indeed inside the cells or just behind the cells, additional spectroscopic studies were performed. Photoluminescence spectra show that the fluorescence intensities of the CdSe/CdS QDs are different between samples at different treatment times. Interestingly, the fluorescence lifetimes are also different. This confirms the interaction between the CdSe/CdS QDs and the intracellular media and that the QDs were successfully seeded into the cells.

**Keywords:** photodynamic properties; time-resolved fluorescence; fluorescence resonance energy transfer

## 1. Introduction

Fluorescent semiconductor nanocrystals or quantum dots (QDs) have become indispensable for biomedical research especially in cellular biology due to their unique photophysical and physicochemical properties. Among the optical properties of QDs that make them attractive for biological applications are size-dependent tunable absorption and emission in the visible and NIR regions, narrow emission bandwidth, broad absorption bands especially for UV excitation, and large single- and multi-photon absorption cross-sections [1–8]. These unique optical properties originate from a combination of bulk semiconductor properties and quantum confinement effect. QDs are also exceptionally bright and photostable compared to fluorescent dyes and proteins [9,10]. Fluorescent dyes usually consist of organic molecules that are easily and irreversibly degraded through photo bleaching by the light used to excite them, progressively emitting less light over time. A multitude of biological applications have exploited these unique properties and to date, QDs have demonstrated utility as probes and labels for biomedical applications, photodynamic agents, in vitro and in vivo fluorophores, and contrast agents for deep-tissue imaging and detection as well as long-term tracking of biological cells, agents for delivery of drugs, molecular-scale platforms for assembling energy transfer-based sensors, theranostic materials, light-harvesting arrays, agents for diagnosis and therapy of genetic disorders and diseases, to name a few [11–21]. The biological application of QDs is still increasing exponentially.

Despite the demonstrated biological applications of QDs, some challenges remain. For instance, cytotoxicity is a major concern when CdSe/CdS QDs are used as agents for drug delivery and disease detection and therapy because although they are carriers, they also have integrated functionalities. On the other hand, the primary challenge when using QDs for cellular imaging is the effective seeding into the cells, since QDs are nanoparticles made of an inorganic core. This difficulty is further compounded by the limited knowledge about the extra- and intracellular processing of such QDs. This is partly due to the strong dependence of the physicochemical and biophysical properties of the nanoparticle when it interacts with a complex biological environment.

Several techniques that enable single particle injection in single cells have been reported [22–30]. For instance, the patch clamp technique inserts a glass micropipette filled with electrolyte into a cell. This technique offers both a high signal-to-noise ratio and temporal resolution [22,23]. Previous works have also reported single particle injection by inserting metal or carbon microelectrodes into the cells, with advantages similar to inserting a glass micropipette [24–26]. However, these techniques require that the micropipettes and microelectrodes are as small as possible to increase the spatial resolution and minimize the invasiveness of the measurement. The overall performance of this technique is further limited by the impedance of the interface between the microstructures and the cell interior, which also limits how small the microstructures can be [22,23]. Recently, three-dimensional (3D) nanostructures such as nanotubes integrated on top of a nanoscale field-effect transistor [27,28] and nanoelectrodes [29] were successfully used to penetrate the cell membrane and enable the delivery of single nanoparticles into single selected cells. The ability to selectively deliver single nanoparticles into single cells benefits many applications. However, these methods are complex and require precise equipment. These are also invasive, with the possibility of disrupting the cell nuclear envelope, which may lead to negative side effects [29]. Endocytosis, which is the passive uptake of nanoparticles, is a non-invasive method of delivering nanoparticles into cells [30]. With this method, the physicochemical properties of the nanoparticles, such as their size, shape, core material, and surface functionalization have a strong impact on cellular interaction, including uptake. Our work uses endocytosis with the assistance of polyethylene glycol (PEG) that enhances the penetration capacity of the nanoparticles, allowing the nanoparticles to trespass into the intracellular cytoplasm. Although single particle injection into single selected cells is not possible through endocytosis, this method is simple, economical, and sufficient for specific applications such as when investigating the photodynamic properties of nanoparticles in intracellular media.

Monitoring the uptake of nanoparticles into cells is important to ensure that the nanoparticles have indeed been injected into the cells. Surface-enhanced Raman spectroscopy (SERS) is a well-known method for this purpose. SERS is a non-destructive and multiplexed technique for live intracellular imaging [31]. Although SERS overcomes the limits inherent in traditional cell imaging techniques, some issues remain. Firstly, it requires high-power lasers or long exposure times to be detected because of the weak scattering from biomolecules. Furthermore, cells usually exhibit a complicated Raman spectrum and interfering background from endogenous biomolecules. This makes target detection and data interpretation very demanding [32]. Fluorescence spectroscopy provides a stronger signal compared to SERS, although it could also have a low signal-to-noise ratio. Previously, Förster Resonant Energy Transfer (FRET) using plasmonic nanostructures has been reported for single molecule detection [33]. The key features of this technique include the reduction of detection volume and improved signal-to-noise ratio. However, the selective functionalization at the specific positions where the plasmonic nanostructure is present is still challenging and not well understood [33].

In this paper, the photodynamic properties of CdSe/CdS QDs in intracellular media are presented. CdSe is among the widely used QDs core due to its size-tunable and stable fluorescence in the visible to NIR regions, the wide availability of precursors, and the well-defined technology of crystal growth [22]. Capping the CdSe core with a CdS shell to produce a CdSe/CdS core/shell QDs offer improved photoluminescence quantum efficiency and red-shifted photoluminescence spectra, making them bright and robust materials for bioimaging [2]. Using fluorescence and time-resolved fluorescence

spectroscopy, we demonstrate the interaction between CdSe/CdS QDs and intracellular media and show that QDs were successfully seeded into Jurkat cells.

## 2. Experimental

### *2.1. Materials*

### 2.1.1. Quantum Dots CdSe/CdS

CdSe/CdS quantum dots were synthesized by the wet chemical method using CdO, Se, and S powders (from Merck). CdO was first dissolved in oleic acid (OA) at 240 °C over 1 h with a CdO:OA ratio of 1:3. Then, ODE was added and the mixture was heated up to 280 °C in order to obtain the complex Cd-OA salt. Next, the Se powder was dissolved in octadecene (ODE) at 180 °C over 5 h and the S powder was dissolved in ODE at 100 °C over 1 h. These were used as precursor solutions. Then, CdSe/CdS quantum dots were synthesized by Cd-OA at 280 °C in a three-neck flask under Nitrogen. Then, the Se precursor was quickly injected into the flask with a Cd-OA:Se ratio of 3:1. Crystalline CdSe/CdS was grown in 5 min. Then, the temperature of the flask was lowered to 240 °C, after which the CdS coating was done. Then, the S precursor was slowly injected into the flask. A CdSe/CdS core/shell with an average size of 5 nm was obtained after 10 min.

### 2.1.2. Bio-Cells

Jurkat cells, the human T lymphoblastoid cell line, were obtained from Dr. Domenico V Delfino, University of Perugia, Perugia, Italy. Fetal bovine serum (FBS), Gentamicin, RPMI-1640, and PEG-1500 were obtained from Sigma Chemical Co. (St. Louis, MO, USA). RPMI-1640 and L-glutamine were purchased from Invitrogen (Carlsbad, CA, USA).

### *2.2. Cell Culture and In Vitro Experiment*

The Jurkat cells were cultured in RPMI-1640 medium supplemented with 10% inactivated fetal bovine serum (FBS) and 50 µg/mL gentamicin at 37 °C and 5% $CO_2$ in a humidified atmosphere. The cell line was seeded at a density of $5 \times 10^4$ cells $mL^{-1}$.

In order to study the uptake capacity of the CdSe/CdS quantum dots, the Jurkat cells at log phase were seeded into 24-well plates at the concentration of $1 \times 10^4$ cells $mL^{-1}$ and $1 \times 10^6$ cells $mL^{-1}$; then, they were incubated for an additional 24 h. Then, polyethylene glycol (PEG-1500) and CdSe/CdS QDs were added to the cell-seeded wells for 3 h and 6 h. PEG interacts with phospholipids, which are the main components of the cell membrane, leading to the cellular membrane loosing and fusing. Thus, in this study, we applied PEG for the purpose of enhancing the penetration capacity of NPs to trespass into the intracellular cytoplasm. At the experimental condition (RPMI-1640 culture medium pH = 7.1, incubation at 37 °C for 3h, 6 h, 9 h and 12 h), we did not observe interaction or absorbance of PEG on the NP surface. After the allotted time, the medium and cells were collected into 15-mL falcon tubes. The tubes were subjected to centrifugation at 1000 rpm for 5 min to separate the cells from the cultured medium. The cells were then triply washed with sterile PBS (phosphate buffer saline) pH = 7. After being resuspended in PBS, the nanoparticle uptake capacity for all treated cells was estimated. Table 1 shows the sample name for the two cell concentrations before treatment.

**Table 1.** Sample note. PEG: polyethylene glycol.

| Sample Name | Cell Concentration (cell/mL) | PEG |
|---|---|---|
| CL | $1 \times 10^4$ | No |
| CL-P | $1 \times 10^4$ | Yes |
| CH | $1 \times 10^6$ | No |
| CH-P | $1 \times 10^6$ | Yes |

### 2.3. Apparatus

The morphology of all the samples was obtained using a Transmission Electron Microscope (TEM—Joel-JEM 1010). The fluorescence microscope images were recorded on Axio Vert.A1 (ZEISS) by using a 532 nm laser excitation source and white light from a xenon lamp. The fluorescence spectra and lifetimes were recorded using the Time-Correlated Single-Photon Counting (TCSPC) method. The purpose-built ps TCSPC setup has an approximately 25 ps time resolution. It uses a diode laser with approximately 0.1 mW output power, approximately 10 ps pulse duration, a 4 MHz repetition rate, a microchannel plate (MCP) photomultiplier tube R3809-50 (Hamamatsu), and an excitation wavelength of $\lambda_{\text{excitation}}$ = 405 nm. The average fluorescence lifetimes were estimated by fitting an exponential function to the decay curves.

## 3. Results and Discussion

### 3.1. Characteristics of CdSe/CdS QDs

The TEM image in Figure 1a shows the sharpness and dispersion of the CdSe/CdS QDs. The QDs are spheres with size ranging from 4 to 6 nm and appear to have monodispersion in the solvent. The unique optical properties of QDs originate from a combination of bulk semiconductor properties and quantum confinement effects. For most materials, dimensions ranging from 2 to 10 nm will satisfy requirements for quantum confinement [20]. The fluorescence spectrum in Figure 1b shows a strong peak at 618 nm. The fluorescence intensity was obtained using a Time-Correlated Single-Photon Counting (TCSPC) method with very high sensitivity as described in Section 2.3. The quantum yield (QY) of the QDs is estimated to be about 0.4. This value of QY is suitable for the labeling and is sufficient for use as target-specific probes at the core of fluorescence signaling, imaging, and sensing.

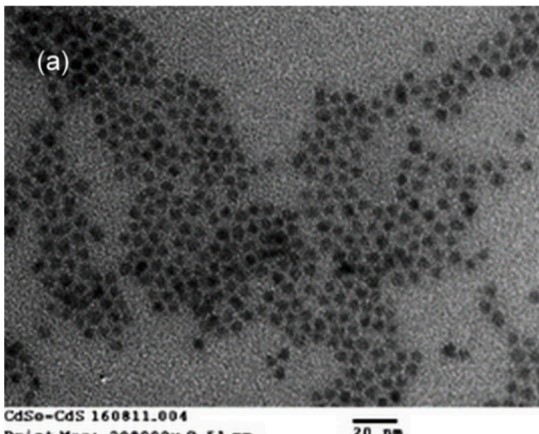 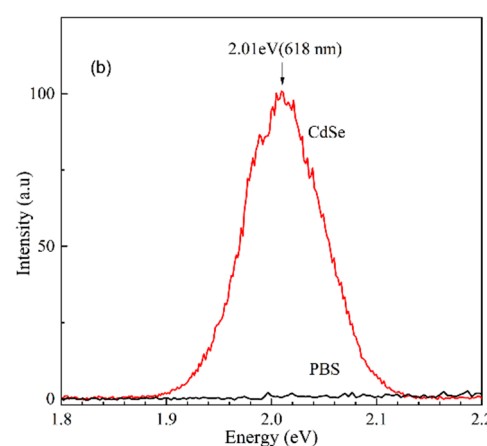

**Figure 1.** Characteristics of the CdSe/CdS quantum dots (QDs). (**a**) TEM image showing the morphology of the CdSe/CdS QDs; (**b**) Fluorescence spectra of the QDs in the solvent obtained with 405 nm excitation showing a fluorescence peak at 618 nm (red). This fluorescence peak is absent in the phosphate buffer saline (PBS) media that is void of QDs (black).

### 3.2. Fluorescence Microscopy Analysis

The fluorescence microscope image of CdSe/CdS QDs in Jurkat cells under 532 nm laser excitation without white light irradiation is shown in Figure 2a. Fluorescence originating from the QDs (red spots) is clearly visible. The red spots are not present in the reference sample containing only the Jurkat cells, which confirms that the observed fluorescence is from the CdSe/CdS QDs. In the presence of white light, it can be observed that areas where fluorescence from the QDs is present appear to be brighter, as shown in Figure 2b. Moreover, the cells appear to be globular with an average diameter of about 11 μm under white light illumination. Despite the difference in the brightness between areas

with and without the QDs, it is difficult to ascertain whether the CdSe/CdS QDs are indeed inside the cells or just behind the cells. Therefore, the photodynamic properties are further investigated using fluorescence spectroscopy.

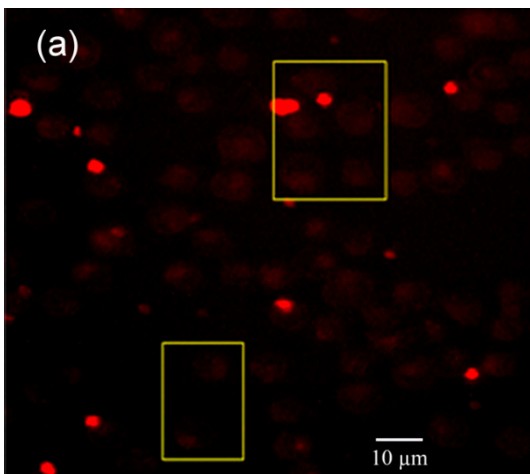 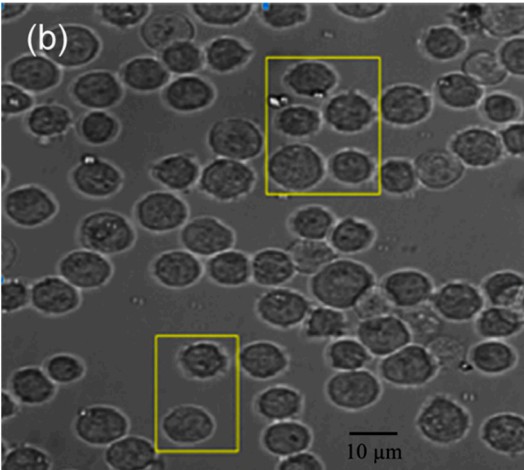

**Figure 2.** Fluorescence microscope images of the CdSe/CdS QDs in Jurkat cells under 532 nm laser excitation: (**a**) without and (**b**) with white light irradiation from a xenon lamp in reflection geometry.

### 3.3. Fluorescence Spectra

Figure 3a–c show the fluorescence spectra of the QDs seeded in the Jurkat cells after 3 h, 6 h, and 9 h of incubation, respectively. The fluorescence spectra of the Jurkat cells without CdSe/CdS QDs are also presented in Figure 3a for comparison. The 618 nm fluorescence peak is characteristic of the CdSe/CdS QDs. This fluorescence peak appeared in samples incubated for 3 h with low cell concentration and treated with PEG ($C_L$-P), low cell concentration without PEG ($C_L$), high cell concentration treated with PEG ($C_H$-P), and high cell concentration without PEG ($C_H$). For the sample incubated for 6 h, the fluorescence peak was only observed after treatment with PEG. This fluorescence peak was not observed in the sample incubated for 9 h and 12 h, even when treated with PEG. This could be because when the incubation time increased, a large number of QDs are inserted into the cells, causing cell death. Subsequently, the QDs were eliminated in the process of washing the sample after incubation. This causes the fluorescence signal of the sample for the 6 h uptake to be lesser than the sample uptake in 3 h. Only one experiment was done for each of the exposure times. The same experimental parameters were used for the samples with 3 h and 6 h exposure times, giving some confidence to the experimental results that indicate the uptake for the 6 h exposure time is less compared to that of the 3 h exposure time. We surmise that more CdSe/CdS QDs have been seeded into the cells when the cell concentration was low.

Therefore, the peak fluorescence intensity is higher in the sample with a low cell concentration compared to the one with a higher cell concentration. Further enhancement in the fluorescence peak intensity is observed when the cells are treated with PEG. When treated with PEG, small holes in the cell membrane would have been enlarged, thereby allowing the QDs to be seeded easily. Several works have reported that polymers such as PEG can improve biocompatibility and facilitate intracellular delivery by driving initial interactions with the plasma membrane and ultimately internalization [34]. Then, the effective seeding resulted in enhanced fluorescence in the PEG-treated cells. As shown in Figure 3a, no fluorescence is observed for the QDs-free Jurkat cells. These results also confirm that the QDs were indeed successfully seeded inside the cells and the fluorescence observed in the microscope images originated from within the cells.

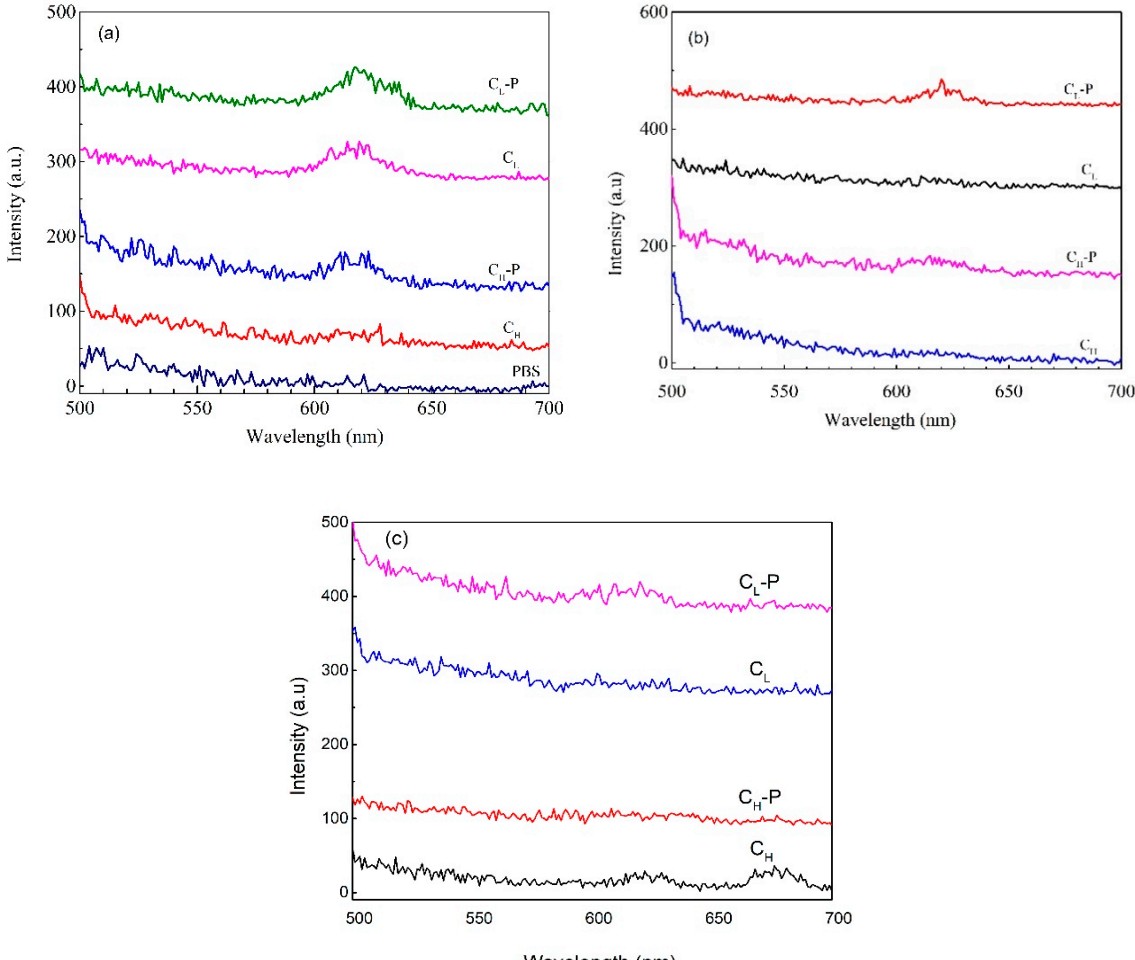

**Figure 3.** Fluorescence spectra of the CdSe/CdS QDs seeded in the Jurkat cells 3 h (**a**), 6 h (**b**), and 9 h (**c**) after treatment. $C_L$: low cell concentration of $1 \times 10^4$ cells mL$^{-1}$; $C_H$: high cells concentration of $1 \times 10^6$ cells mL$^{-1}$; $C_L$-P; $C_H$-P: polyethylene glycol (PEG)-treated cells. The fluorescence spectra of the Jurkat cells without CdSe/CdS QDs are also presented in (**a**) for comparison.

*3.4. Fluorescence Lifetime*

Figure 4 shows the fluorescence lifetime of the CdSe/CdS QDs in different intracellular media, namely low cell concentration treated with PEG ($C_L$-P), low cell concentration without PEG ($C_L$), and high cell concentration treated with PEG ($C_H$-P). Fluorescence from the high cell concentration sample without PEG treatment is too weak to yield any fluorescence lifetime measurement. For comparison, the fluorescence lifetime of the QDs in just the solvent is also obtained. It can be observed that the fluorescence decay curves have a fast and slow component as it initially decreases quickly and then slowly. Therefore, the fluorescence decay curves were fitted to a biexponential function to have two lifetime values. A least squares fit was used to ensure that the correct function was utilized. The fit has an R2 value of 0.994. This value suggests a good agreement between the experimental curve and the data fitting curve. The first lifetime is less than 1.2 ns and varies with the samples, while the second lifetime is 2 ns and does not change with the samples. The resolution of the TCSPC set-up used to obtain the fluorescence lifetime is approximately 35 ps and the pulse duration of the laser used is approximately 50 ps. On the other hand, the fast fluoresce decay time is 1.2 ns; therefore, the measured fluorescence lifetimes are not biased by limitations to the set-up. These fluorescence lifetimes can be attributed to surface recombination and deep trap recombination, respectively. Fluorescence lifetimes due to surface recombination are summarized in Table 2. The results show that the fluorescence lifetime is faster, almost decreasing to half when the QDs were placed in the intracellular media

compared to when they were in the water. The fluorescence lifetime when the QDs are in the solvent is 1.92 ns, whereas the lifetime in the intracellular media ranged from 0.93 to 1.21 ns, depending on the cell concentration. The decrease in fluorescence lifetime can be attributed to the removal of the QDs' surfactant ligands, which are then replaced with small molecules such as amines that are present in the intracellular media [34–37]. They may also be caused by fluorescence resonance energy transfer [38–40]. In this case, the binding protein serves as the acceptors, while the QDs are the donor. The QDs acceptors reabsorb the resonance energy from the protein donor, thereby resulting in the direct change in the fluorescence lifetime of the donor moiety.

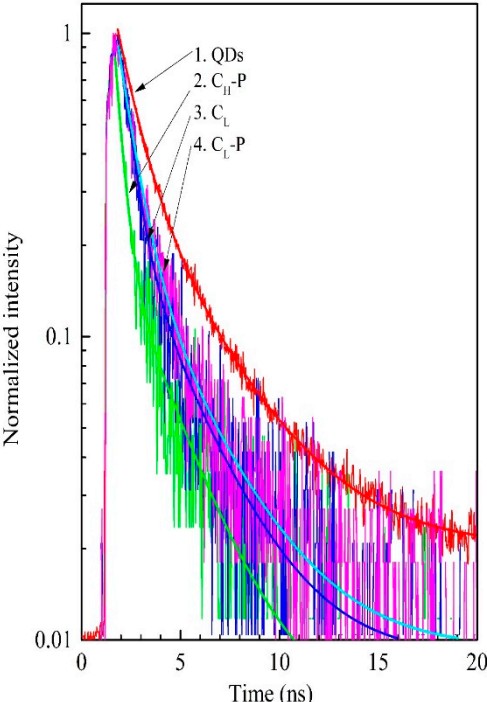

**Figure 4.** Fluorescence lifetime of CdSe/CdS in intracellular media for different cell concentrations. $C_L$: low cell concentration of $1 \times 10^4$ cells mL$^{-1}$; $C_H$: high cell concentration of $1 \times 10^6$ cells mL$^{-1}$; $C_L$-P; $C_H$-P: PEG-treated cells. The fluorescence lifetime when the QDs are in the water is also presented for comparison (QDs). The fluorescence lifetime for the high cell concentration sample is not shown, as the fluorescence is too weak.

**Table 2.** Fluorescence lifetime of the QDs in the cell and in water.

| Sample | $\tau$ (ns) |
|:---:|:---:|
| QDs in water | 1.92 |
| $C_L$ | 1.06 |
| $C_L$-P | 1.21 |
| $C_H$ | // |
| $C_H$-P | 0.93 |

## 4. Conclusions

In summary, we have described the preparation and seeding of CdSe/CdS QDs into Jurkat cells. The fluorescence microscope image and photodynamic characteristics suggest that the QDs were successfully seeded into the cells. Fluorescence intensity is observed to increase in the sample with a low cell concentration as more QDs are seeded into a single cell. Further enhancement in fluorescence intensity is observed when the cells are treated with PEG, which confirms facilitated intracellular delivery. The fluorescence lifetime also changes when the QDs are seeded into the cell. This can

be attributed to the removal of the surfactant ligands of the QDs or fluorescence resonance energy transfer in intracellular media. The change in the fluorescence lifetime demonstrates that there was an interaction between the QDs and the intracellular media. Measurement of the fluorescence lifetime can be a valuable tool for investigating interactions between the CdSe/CdS QDs and the intracellular media and can provide clues as to whether the CdSe/CdS QDs are indeed inside the cells.

**Author Contributions:** Conceptualization, methodology, T.B.N., T.B.V., and T.T.D.; software, validation, formal analysis, investigation, resources, data curation, T.B.N., D.C.N., H.M.P., and T.T.D.; writing—original draft preparation, T.B.N., and T.B.V.; writing—review and editing, T.B.N., and M.C.-R.; project administration, T.B.V. All authors have read and agreed to the published version of the manuscript.

**Funding:** This research was funded by the National Foundation for Science and Technology Development of Vietnam (NAFOSTED) Grant number ĐT.NCCB-ĐHUD.2012-G/07.

**Conflicts of Interest:** The authors declare no conflict of interest.

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
