# Peer review of "Photodynamic Properties of CdSe/CdS Quantum Dots in Intracellular Media"

_applsci, doi:10.3390/app10113988_

Round 1

Reviewer 1 Report

The Authors should provide an extensive comparison with state-of-the-art techniques enabling single particle injection in single cells (see for instance Huang et al. Nano Lett. 2019, 19, 2, 722-731 or Caprettini et al. Adv. Sci. 2018, 5, 12, 1800560). Moreover, it is somewhat missing the advantage to use quantum dots for fluorescence emission enhancement compared to other techniques based for instance on plasmonic platforms (see for instance Ardini et al. Sci. Rep. 2018, 8, 12652 or Zambrana-Puyalto et al. Nanoscale Adv. 2019, 1, 2454-2461).

The spectra in Figure 3 are not quite reliable: what is the cut-off threshold the Authors apply to claim that they are detecting a signal? It is quite hard to discriminate some peaks from the fluorescence spectra they are showing. I suggest them to increase the particle concentration to ensure that they have enough signal to support their claim. Same comment can be applied to Figure 4...the noise is to high to discriminate the effect of using different cells concentrations. Moreover, it is not quite clear why the Authors change the cells concentration rather thatn the QDs concentration, which to me makes more sense.

Author Response

We would like to thank the reviewer for providing insight that has helped us revise and improve the paper. We have made the following changes following the reviewer's suggestions:

Reviewer 1

  • The Authors should provide an extensive comparison with state-of-the-art techniques enabling single particle injection in single cells (see for instance Huang et al. Nano Lett. 2019, 19, 2, 722-731 or Caprettini et al. Adv. Sci. 2018, 5, 12, 1800560). Moreover, it is somewhat missing the advantage to use quantum dots for fluorescence emission enhancement compared to other techniques based for instance on plasmonic platforms (see for instance Ardini et al. Sci. Rep. 2018, 8, 12652 or Zambrana-Puyalto et al. Nanoscale Adv. 2019, 1, 2454-2461).

Answer: Our study is different from others since we did not inject our NPs into cells. Instead, we let the leukemic cells absorb the NPs passively. To enhance the penetrating process of NPs, we used polyethylene glycol (PEG) which were reported to interact with the phospholipids of a cell membrane, leading to the cellular membrane loosing and fusing. Moreover, the experiments were conducted on Jurkat cells, human acute T cell leukemias cultured in suspension form, which can help us to look at the NPs’ interaction with blood cells in the human body.

Fluorescence and Raman spectroscopy are two completely different techniques, the fluorescence spectra give a much greater signal than the Raman spectrum despite the surface-enhanced Raman scattering (SERS) so we do not compare these two techniques.

  • The spectra in Figure 3 are not quite reliable: what is the cut-off threshold the Authors apply to claim that they are detecting a signal? It is quite hard to discriminate some peaks from the fluorescence spectra they are showing. I suggest them to increase the particle concentration to ensure that they have enough signal to support their claim. Same comment can be applied to Figure 4...the noise is to high to discriminate the effect of using different cells concentrations. Moreover, it is not quite clear why the Authors change the cells concentration rather that the QDs concentration, which to me makes more sense.

Answer: The peak at 618 nm is quite obvious in Figure 3. We confirmed that this 618 nm peak is the fluorescence emission from CdSe / CdS QD by testing the Jurkat cel and PBS medium without the CdSe/CdS QDs. With just the reference Jurkat cel and PBS medium, this 618 nm peak was absent and we did not observe fluorescence emission in this wavelength range. The spectra shown in Figure 3 were measured on the system by fluorescence using TCSPC technique with very high sensitivity as described in section 2.3.

We have also sought to introduce more QDs into the cell to increase the fluorescence signal such as increasing the QD concentration, increasing time cell-seeded wells for 9 hours, and 12 hours. Samples were obtained after centrifuging, washing, and observing under a microscope to ensure that cells were destroyed causing QDs toxicity.

Reviewer 2 Report

CdSe/CdS QD are highly toxic substances, because of the possibility to release Cd2+ ions, and the nanoparticle (NP) photocatalytic activity. There are many works in the literature dealing with the toxicity of these materials (DOI: 10.1016/j.scitotenv.2018.04.206, 10.1177/1179670717694523,  10.1002/smll.200600218) and there are many studies which propose to avoid these adverse effects with the use of suitable protective shells formed by, for example, ZnS, glass or polymer (DOI: 10.1177/1179670717694523, 10.1021/nl047996m) for this reason it is not clear to me why the author want to test this kind of nanoparticles as possible luminescent labels in the biomedical field.  

This is the first and most important issue I have with respect to the manuscript. Why characterize and test NP for a use for which they are not suited?. Besides this, there are other critical points that should be addressed before the article can be submitted again for publication:

1) In the whole paper there is no statistic done on the experiments performed: how many sample were tested?. Did the authors repeat the exposition experiment more than one time, in order to verify that indeed uptake in the 6 h exposure is lesser than in the 3 hour exposure? A sound statistical analysis in these experiments is necessary to grant publication.

2) In Figure 2 the picture recorded with white light is just an image of the sample in reflection? If so the authors should clarify it.

3) How were the Jurkat cells deposited on the microscope slide?.

3) Are the author really sure that centrifugation will remove the CdSe/CdS nanoparticles that stick on the cell surface, but are not internalized? In honesty I have to admit that this is something that all authors in the filed do and give for granted, to my knowledge.

4) The role of PEG is not clear to me, and I think that first of all it is important to assess how it affects the fluorescence properties (intensity and lifetime)  of the CdSe/CdS NP when dissolved in PBS without the Jurkat cells. Can the PEG be adsorbed on the NP surface?

5) Since the fluorescence decay curves presented in Figure 4 are clearly multiexponential decay. How were calculated the decay times reported in Table 2 and what is the error affecting these measurements?

6) The final comment on quenching of the quantum dots fluorescence by FRET is just confusing and in my opinion completely wrong. What are the binding proteins? How do the author justify that the  “binding protein” is excited at 405 nm, since proteins do not absorb in the visible?. If the QD act as acceptor, their lifetime should not be reduced, what can happen is: (i) a raise in fluorescence with the typical decay time of the donor when the decay time of the donor is shorter than the one of the acceptor, or (ii) a lengthening of the decay time of the acceptor, if the decay time of the donor is longer than the one of the acceptor.

Author Response

We would like to thank the reviewer for providing insight that has helped us revise and improve the paper. We have made the following changes following the reviewer's suggestions:

CdSe/CdS QD are highly toxic substances, because of the possibility to release Cd2+ ions, and the nanoparticle (NP) photocatalytic activity. There are many works in the literature dealing with the toxicity of these materials (DOI:

10.1016/j.scitotenv.2018.04.206,10.1177/1179670717694523,  10.1002/smll.200600218) and there are many studies which propose to avoid these adverse effects with the use of suitable protective shells formed by, for example, ZnS, glass or polymer (DOI: 10.1177/1179670717694523, 10.1021/nl047996m) for this reason it is not clear to me why the author want to test this kind of nanoparticles as possible luminescent labels in the biomedical field. 

This is the first and most important issue I have with respect to the manuscript. Why characterize and test NP for a use for which they are not suited?. Besides this, there are other critical points that should be addressed before the article can be submitted again for publication:

Answer: We also know that quantum dots that use cadmium is highly toxic, but in some experiments in the biological environment, they still use these quantum dots because of their high quantum yield compared to other quantum dots. In our study, the toxicity of CdSe/CdS quantum dots were also tested at different concentrations so that the sample could be maintained for 24 h without destroying the cells.

(1) In the whole paper there is no statistic done on the experiments performed: how many sample were tested?. Did the authors repeat the exposition experiment more than one time, in order to verify that indeed uptake in the 6 h exposure is lesser than in the 3 hour exposure? A sound statistical analysis in these experiments is necessary to grant publication.

Answer: In this paper, we present results of 2 series of samples, one with an uptake in 3 hours and another in 6 hours but we have also done experiments with many sample series having different quantum dot concentrations and uptake times of 9 hours and 12 hours. The samples with high QDs concentration and high uptake time (9 hours, 12 hours) after centrifuging, washing, and observing under a microscope that cells were destroyed showed no photoluminescence. This could be because when large amounts of QDs are inserted into the cell, the cells were destroyed and the QDs were lost in the process of eliminating the sample.  This causes the fluorescence signal of the sample for the 6 h uptake to be  lesser than the sample uptake in 3 h.

(2) In Figure 2 the picture recorded with white light is just an image of the sample in reflection? If so the authors should clarify it.

Answer: In the caption of figure 2, we have written that (a) is the fluorescence microscope image under 532 nm laser excitation and (b) is microscope image (reflection – without laser excitation)  under xenon lamp irradiation

(3) How were the Jurkat cells deposited on the microscope slide?.

Answer: As mentioned in the method part, nanosample incubated cells were washed with phosphate buffer saline (PBS) three times. After the final centrifugation, the supernatant were removed. The obtained Jurkat cell pellets were resuspended in 200 µL PBS and directly spreaded over on the clean surface of glass slides, covered with cover slip and immediately observed under microscope.

(4) Are the author really sure that centrifugation will remove the CdSe/CdS nanoparticles that stick on the cell surface, but are not internalized? In honesty I have to admit that this is something that all authors in the filed do and give for granted, to my knowledge.

Answer: The purpose of centrifuging and washing is to eliminate extracellular quantum dots (on cell membranes and in PBS solvents) that retain QDs only in the cell but practically  it is impossible to confirm if we have completely removed them. Due to the limitations of optical microscopy, we could not clearly visualize the QDs (nano size). Therefore, we have to measure the time resolution spectrum to confirm that the QDs have been injected in the cells.

(5) The role of PEG is not clear to me, and I think that first of all it is important to assess how it affects the fluorescence properties (intensity and lifetime)  of the CdSe/CdS NP when dissolved in PBS without the Jurkat cells. Can the PEG be adsorbed on the NP surface?

Answer: As reported, PEG interacts with phospholipids, which are the main components of cell membrane, leading to the cellular membrane loosing and fusing. Thus, in this study, we applied PEG for the purpose of enhancing the penetration capacity of NPs to trespass into the intracellular cytoplasm. At the experimental condition (RPMI-1640 culure medium pH=7.1, incubation at 37 oC for 3h, 6h, 9h), we did not observe the interaction/absorbance of PEG on the NP surface. We added this information in page 3, line 85.

(6) Since the fluorescence decay curves presented in Figure 4 are clearly multiexponential decay. How were calculated the decay times reported in Table 2 and what is the error affecting these measurements?

Answer: Figure 4 shows that the fluorescence decay curves decrease quickly and then decrease slowly. Therefore, the fluorescence decay curves were fitted to the biexponential function to have two-lifetime values, where the first lifetime with a value of fewer than 1.2 ns and varies with the samples and the second lifetime has a value of 2 ns that does not change with the samples. These fluorescence lifetimes can be attributed to the surface recombination and deep trap recombination. The error depends on the spectral signal magnitude, in our case the intensity of the deccay curve is large enough so the fluorescence lifetime fitting value is reliable.

(7) The final comment on quenching of the quantum dots fluorescence by FRET is just confusing and in my opinion completely wrong. What are the binding proteins? How do the author justify that the  “binding protein” is excited at 405 nm, since proteins do not absorb in the visible?. If the QD act as acceptor, their lifetime should not be reduced, what can happen is: (i) a raise in fluorescence with the typical decay time of the donor when the decay time of the donor is shorter than the one of the acceptor, or (ii) a lengthening of the decay time of the acceptor, if the decay time of the donor is longer than the one of the acceptor.

We considered that the change in the surface state of QDs in cell intracellular is the main cause of the reduction of fluorescence lifetime. However, FRET can also be the cause of this change, like some published [26-28]. Here we have confused the role of the quantum dot gate and have corrected it in the manuscript

Round 2

Reviewer 1 Report

The Authors have not tried to compare their approach with others I suggested, which is a pity because it can give a more clear contextualization of their work compared to other techniques. This again is quite annoying since if the work cannot be well contextualized, to make comparison with alternative techniques (see the references I suggested to the Authors in my previous report) is difficult and so the present work might not be of any interest if the reader is not able to contextualize it.

Passive absorption of nanoparticles by cells is pretty standard (so I do not understand the emphasis of the Authors in claiming that "our approach is different"). See, in addition to the works already suggested, works by Bianxiao Cui, Bozhi Tian, Molly Stevens, Francesco De Angelis, etc.

I agree with the Authors that that fluorescence is stronger than Raman, but at the same is more noisy and problematic to deal with (this is why I suggested to try to make a comparison with Raman and also plasmonic approaches which are clearly superior and the actual state-of-the art).

Author Response

We thank for pointing out various techniques that contextualizes our work. We apologize for not explicitly making this clear and citing the references provided by the reviewer in the first revision. We have now added an extensive review and comparison of our work with other techniques. We have also cited all the references suggested by the reviewer. These are cited as references 22-34 in the revised manuscript and enumerated below:

  1. Sakmann, B. & Neher, E. Patch clamp techniques for studying ionic channels in excitable membranes. Rev. Physiol., 1984, 46, 455–472.
  2. Areles Molleman, Patch Clamping: An Introductory Guide to Patch Clamp Electrophysiology,Wiley, 2003.
  3. Hai A., Shappir J., & Spira M. E., In-cell recordings by extracellular microelectrodes. Nature Methods 2010, 7(3), 200–202.
  4. Schrlau, M. G., Dun, N. J. & Bau, H. H. Cell electrophysiology with carbon nanopipettes. ACS Nano 2009, 3, 563–568.
  5. De Asis E. D., Leung J., Wood S., & Nguyen C. V., High spatial resolution single multiwalled carbon nanotube electrode for stimulation, recording, and whole cell voltage clamping of electrically active cells. Phys. Lett. 2009 95, 153701.
  6. Jian-An Huang, Valeria Caprettini, Yingqi Zhao, Giovanni Melle, Nicolò Maccaferri, Lieselot Deleye, Xavier Zambrana-Puyalto, Matteo Ardini, Francesco Tantussi, Michele Dipalo, and Francesco De Angelis, On-Demand Intracellular Delivery of Single Particles in Single Cells by 3D Hollow Nanoelectrodes, Nano Lett. 2019, 19, 722−731.
  7. https://www.nature.com/articles/nnano.2011.223#Bib1
  8. https://pubs.acs.org/doi/pdf/10.1021/acs.nanolett.8b03764
  9. https://onlinelibrary.wiley.com/doi/epdf/10.1002/advs.201800560
  10. Sean D. Conner & Sandra L. Schmid, Regulated portals of entry into the cell, Nature, 2003, 422, 38-44
  11. https://www.nature.com/articles/s41598-018-31165-3
  12. Eberhardt, K., Stiebing, C., Matthäus, C., Schmitt, M. & Popp, J. Advantages and limitations of Raman spectroscopy for molecular diagnostics: an update. Expert Rev. Mol. Diagn. 2015, 15, 773–787.
  13. Xavier Zambrana-Puyalto, a Nicolo Maccaferri, Paolo Ponzellini, Giorgia Giovannini, Francesco De Angelisa and Denis Garoli, Site-selective functionalization of plasmonic nanopores for enhanced fluorescence emission rate and Forster resonance energy transfer, Nanoscale Adv., 2019, 1, 2454.

In order to give a comprehensive review of current techniques, the following sentences were also specifically added in the Introduction page 2, starting from line 53.:

“Several techniques that enable single particle injection in single cells have been reported [22-30]. For instance, the patch clamp technique inserts a glass micropipette filled with electrolyte into a cell. This technique offers both high signal-to-noise ratio and temporal resolution [22, 23]. Previous works have also reported single particle injection by inserting metal or carbon microelectrodes into the cells, with advantages similar to inserting a glass micropipette [24-26]. However, these techniques require that the micropipettes and microelectrodes are as small as possible to increase the spatial resolution and minimize the invasiveness of the measurement. The overall performance of this technique is further limited by the impedance of the interface between the microstructures and the cell interior, which also limits how small the microstructures can be [22, 23]. Recently, three-dimensional (3D) nanostructures such as nanotubes integrated on top of a nanoscale field-effect transistor [27,28] and nanoelectrodes [29, 30] were successfully used to penetrate the cell membrane and enable delivery of single nanoparticles into single selected cells. The ability to selectively deliver single nanoparticles into single cells benefit many applications. However, these methods are complex and require precise equipment. These are also invasive, with the possibility of disrupting the cell nuclear envelope which may lead to negative side effects [30]. Endocytosis, which is the passive uptake of nanoparticles is a non-invasive method of delivering nanoparticles into cells [31]. With this method, the physicochemical properties of the nanoparticles, such as their size, shape, core material and surface functionalization have a strong impact on cellular interaction including uptake. Our work uses endocytosis with the assistance of polyethylene glycol (PEG) that enhances the penetration capacity of the nanoparticles, allowing the nanoparticles to trespass into the intracellular cytoplasm. Although single particle injection into single selected cells is not possible through endocytosis, this method is simple, economical, and sufficient for specific applications such as when investigating the photo-dynamic properties of nanoparticles in intra-cellular media. 

Monitoring the uptake of nanoparticles into cells is important to ensure that the nanoparticles have indeed been injected into the cells. Surface enhanced Raman spectroscopy (SERS) is a well-known method for this purpose. SERS is a non-destructive and multiplexed technique for live intracellular imaging [32]. Although SERS overcomes the limits inherent in traditional cell imaging techniques, some issues remain. Firstly, it requires high power lasers or long exposure times to be detected because of the weak scattering from biomolecules. Furthermore, cells usually exhibit a complicated Raman spectrum and interfering background from endogenous biomolecules. This makes target detection and data interpretation very demanding [33]. Fluorescence spectroscopy provides a stronger signal compared to SERS, although it could also have a low signal-to-noise ratio. Previously, Förster Resonant Energy Transfer (FRET) using plasmonic nanostructures have been reported for single molecule detection [34]. The key features of this technique include the reduction of detection volume and improved signal-to-noise ratio. However, the selective functionalization at the specific positions where the plasmonic nanostructure is present is still challenging and not well-understood [34].”

Respectfully yours,

Prof. Vu Thi Bich

Institute of Theoretical and Applied Research, Duy Tan University

Hanoi 100000, Vietnam;

Reviewer 2 Report

in may first referee report on point 1) I asked the authors to add the statistic of the experiments to the experimental data reported and in particular how many independent experiments where done at 3h and 6h exposure. In the answer they reply that they did experiments at 9h and 12h, and do not add any error bar or confidence interval to the data of fluorescence intensity or lifetime reported in the paper. Because of this I do not deem the present paper to be viable for publication.

Author Response

We would like to thank for providing insight that has helped us revise and improve the paper. We have revised the manuscript following all the comments and suggestions of the reviewers, as detailed below:

Unfortunately, it is not possible to repeat the experiment this time because of time constraints, where additional samples need to be ordered and prepared before the actual experiment can be performed. At the time the experiment was performed, only one experiment was done for each of the exposure times. The same experimental parameters were used for the samples with 3h and 6h exposure times, giving some confidence to the experimental results that the uptake for the 6 h exposure time is less compared to that of the 3 h exposure time. The fluorescence intensity was obtained using a Time-Correlated Single-Photon Counting (TCSPC) method with very high sensitivity as described in section 2.3. Likewise, the fluorescence lifetime was obtained using the same method. The resolution of the TCSPC set-up is ~25ps and the pulse duration of the laser used is ~10ps. On the other hand, the fast fluoresce decay time is 1.2 ns, therefore we are confident that the measured fluorescence lifetime graphs are not biased by limitations to the set-up. The fast and slow fluorescence decay components were obtained using a biexponential curve fit to the fluorescence lifetime graphs. We used the least squares fit to ensure the correct function was utilized. The fit has an R2 value of 0.994.

The following sentences were added to the manuscript to clarify these sources of error in the experiment:

Page 4, line 151: “The fluorescence intensity was obtained using a Time-Correlated Single-Photon Counting (TCSPC) method with very high sensitivity as described in section 2.3.

Page 6, line 184: “Only one experiment was done for each of the exposure times. The same experimental parameters were used for the samples with 3h and 6h exposure times, giving some confidence to the experimental results that the uptake for the 6 h exposure time is less compared to that of the 3 h exposure time.”

Page 8, from line 238: “A least squares fit was used to ensure that the correct function was utilized. The fit has an R2 value of 0.994. This value suggests a good agreement between the experimental curve and the data fitting curve. The first lifetime is less than 1.2 ns and varies with the samples while the second lifetime is 2 ns and does not change with the samples. The resolution of the TCSPC set-up used to obtain the fluorescence lifetime is ~25ps and the pulse duration of the laser used is ~10ps. On the other hand, the fast fluoresce decay time is 1.2 ns, therefore the measured fluorescence lifetimes are not biased by limitations to the set-up.”

Furthermore, additional details about the different incubation times have been added in page 6, starting from line 173, as follows:

“Figures 3a, 3b and 3c show the fluorescence spectra of the QDs seeded in the Jurkat cells after 3 h, 6 h and 9 h of incubation, respectively. The fluorescence spectra of the Jurkat cells without CdSe/CdS QDs are also presented in Figure 3a for comparison. The 618 nm fluorescence peak is characteristic of the CdSe/CdS QDs. This fluorescence peak appeared in samples incubated for 3 h with low cell concentration and treated with PEG (CL-P), low cell concentration without PEG (CL), high cell concentration treated with PEG (CH-P), and high cell concentration without PEG (CH). For the sample incubated for 6 h, the fluorescence peak was only observed after treatment with PEG. This fluorescence peak was not observed in the sample incubated for 9h and 12h, even when treated with PEG. This could be because when the incubation time increased, a large number of QDs are inserted into the cells, causing cell death. Subsequently, the QDs were eliminated in the process of washing the sample after incubation. This causes the fluorescence signal of the sample for the 6 h uptake to be lesser than the sample uptake in 3 h.” 

Respectfully yours,

Prof. Vu Thi Bich

Institute of Theoretical and Applied Research, Duy Tan University

Hanoi 100000, Vietnam;

Round 3

Reviewer 1 Report

The Authors have now made a nice discussion which helps the reader to contextualize their work. For me now the article is suitable for publication.

Just few additional minor comments. In Ref. 34 some of the authors' names are not correctly spelled. In other references (e.g. 28, 29, 30, 32) the Authors have put only the link or the DOI...please write the references in a homogeneous manner with authors' names, name of the journal, volume, page and year.

Reviewer 2 Report

I appreciate the honesty of the authors about the number of replica of the experiments they did. Clearly this limit the value of the work itself. On the other hand the paper is still offering important results. For this reason I suggest to publish it as it is, although I still think that the shortening of the nanoparticle lifetime when inside the cell is indeed caused not by energy transfer to the proteins, but to a change of the trap state concentration on the NP surface when the original surfactant is disced by proteins of the cell growth medium or the amines that the authors are recalling.